# Forces Causing One-Millimeter Displacement of Bone Fragments of Condylar Base Fractures of the Mandible after Fixation by All Available Plate Designs

**DOI:** 10.3390/ma12193122

**Published:** 2019-09-25

**Authors:** Marcin Kozakiewicz, Rafal Zieliński, Michał Krasowski, Jakub Okulski

**Affiliations:** 1Department of Maxillofacial Surgery, Medical University of Lodz, 1st Gen. J. Hallera Pl., 90-647 Lodz, Poland; qed@op.pl (M.K.); jakub.okulski@gmail.com (J.O.); 2Material Science Laboratory, Medical University of Lodz, 251st Pomorska, 92-213 Lodz, Poland; michal.krasowski@gmail.com

**Keywords:** mandible, condyle base fracture, plate rigid fixation

## Abstract

Background: There has been no direct comparison of all existing plates dedicated for fracture osteosynthesis of mandibular condyle base until now. The aim of the study was to test mechanically all available designs of titanium plates on the market on polyurethane mandibles using an individually designed clamping system. Methods: Forces required for a 1 mm displacement of fixed fracture and incidents of screw loosening were recorded. Results indicated the best mechanical plates among all existing designs available. Results: It has occured that some of osseofixation plates should not be used any more, whereas some shape of the single plates are similar shape to two single plates shape are regarded as the best osseofixation method for condyle base fracture. Conclusion: General observation is the bigger plate and more screws, the better rigid stable osteosynthesis of mandibular condyle base. 4 plates of current designs of total 30 tested series can be recommended for open rigid internal fixation of fractures of the base of the mandibular condyle. The rest of 26 existing plates should not be used in condylar base fractures.

## 1. Introduction

### 1.1. Epidemiological Information

The mandible is the most vulnerable bone to fractures in the maxillofacial complex. In Europe, mandibular fractures amounted to 42% of all maxillofacial fractures in a recent prevalence study. Condyle extra-capsular fractures consisted of 26% of mandibular fractures in the same previously mentioned European study, ranking first among all types of mandibular fractures [1]. The mandibular condyle or subcondylar region is one of the most common sites of mandibular fracture encountered between 25% and 35% of all mandibular fractures [2,3].

### 1.2. Surgical Procedures

Surgical treatment is performed under general anaesthesia. Generally maxillofacial surgeons use three different surgical approaches to reach the fracture in the region of the condylar process of the mandible. For cases where a submandibular approach is necessary for load bearing osteosynthesis of associated mandibular body fractures such as in cases of atrophic mandibles the already necessitated submandibular approach was used for open reduction and internal fixation of condylar process fracture. In cases of mild and moderately displaced condylar base fractures, a transoralendoscopically assisted approach is chosen. Condylar process fractures are reached by retromandibular transparotid approaches.

### 1.3. Ossoefixation Plates

The main aim of the study was to compare all known condylar plates. Researchers of this study assumed that unavailability a particular plate on the market does not mean the plate was worse than others. Thus, the authors compared all known original plates including those that are hard to obtain.

Plenty of plates dedicated for mandibular condyle fracture fixation is available. A little help in information is literature selection of the most proper plate for clinical situations. There are only fragmentary studies concerning one or few plates [4,5,6,7,8,9,10,11,12,13,14]. Open rigid internal fixation is the standard surgical procedure when the fracture is dislocated or significantly displaced.

In the literature, bicortical screws [15], microplates [15], a single titanium screw or pin [16], two resorbable screws [17], and resorbable pins [18] have been used for rigid osseofixation.

Elimination of cut micromovements from fracture line is the most important for uneventful fracture healing after plate fixation [14]. It requires proper plate stably fixed by screws. Then fracture line movement can be limited to the value significantly less than 1 mm [4,12]. The main question is which of many plates available on the market is the best for rigid fixation of a basal fracture of mandibular condyle?

Aim of this study was the mechanical comparison of 30 plate designs dedicated for fracture osteosynthesis of mandibular condyle base.

## 2. Material and Methods

### 2.1. Mandibles

Solid polyurethane foam mandibles were utilized in this study (Figure 1). The high variability in the density and the elastic modulus of bone affects biomechanical testing results [19]. Synthetic foam materials have been shown to produce less intra- and inter-specimen variability than cadaver bone [20]. A foam block has consistent material properties, similar to the human cancellous bone. Solid polyurethane foam is widely used as an ideal medium to mimic human cancellous bone and has been confirmed by the American Society for Testing and Materials [21,22] as a standard material for testing orthopedic devices and instruments. In this study, polyurethane foam (Sawbones, Vashon, WA, USA: density 0.16 g/cc, compression modulus 58 MPa) was used as a substitute for bone [23,24,25,26].

### 2.2. Plates

There was collected information of all available dedicated plates for rigid fixation of condylar process of mandible (Table 1). Next, similar plates were laser cut from medical certified titanium sheet (grade 23, 1-millimeter thickness).

The condylar base were cut in level of typical basal condylar fracture in each model. Subsequently, proximal (i.e., condylar) and distal (i.e., ramus, mandibular) fracture segment were fixed by plate and the same 6-mm length self-tapping screws of 2.0 system. Predrilling was made by 1.5 mm drill. Each hole in the plates were filled by screws. 7 models were tested for each plate design.

### 2.3. Simulation Set

The condyles were set at a 15° inferior tilt in the sagittal plane and at a 10° lateral in the coronal plane to simulate actual masticatory force loading on the temporomandibularjoint. This model results in the condyle exerting a force upwards and some what forwards and medially [6].

For testing purposes Zwick Roell Z020 universal strength machine (Zwick-Roell, Ulm, Germany) with individually-made clamping system was used. Clamping system comprised flat 1 mm thick stainless steel based on 70 cm × 60 cm angulated aluminum block with milled 4 × M6 threaded holes for screwing flat base plate (Figure 2). On the plate for stabilization of mandibule stainless steel try square was used. Pre-load force was 1N and test speed was 1 mm/min. The action point of the compressive forces was located at the condyle. The load vs displacement relationship, load for permanent deformation, and maximum load at fracture were recorded using the lnstron chart recorder. Permanent deformation was defined as the initial point that the load-displacement relationship was no longer linear. Maximum load was defined as the greatest load recorded just before any sudden decrease in load level (Figure 3).

### 2.4. Statistical Analysis

Number of holes in the plate, plate height, plate width surface area of the plate is faced to the bone were noted for interpretation of the experimental data.

Statistical analysis was performed in Statgraphics Centurion 18 (Statgraphics Technologies Inc. The Plains, VA, USA). Kruskal–Wallis test was applied for between design comparison. Categorical variables were tested for independence by Chi-Square test. Objective description of plate designs was attempted basing on factor analysis due to the need of indicating the best plates. The mathematical purpose of the analysis was to obtain a small number of factors which account for most of the variability in the 4 bases variables describing plate features: height (mm), width (mm), plate surface area (mm^2^), total fixing screws. Neural Network Bayesian Classifier, i.e., a probabilistic neural network (PNN) was used to classify designs into different condyle screw pullout based on 4 input variables of the 210 mechanical tests: Plate Design Factor, Fmax/dL (N/mm), number of screws in condyle, number of screws in ramus. PNN had 2 hidden layers, and two out puts: with and without pulled out screw from condyle fragment.

### 2.5. Surface Treatment

Surface treatments of metals are intended to produce a biologically active surface. Different macrosurface designs have influence to the tissues’ retention. In order to obtain an ideal, efficient roughness as already demonstrated different techniques have been proposed. The implant surfaces are subdivided into two large groups, smooth and rough.

Rough surface scan be obtained with two types of treatments, additive and subtractive techniques. Additive techniques include the following:
Titanium Plasma SprayCoating with hydroxyapatiteAnodic oxidation

Subtractive techniques include the following:
Sandblasting with alumina oxideSandblasting with titanium particlesSandblasting with soluble or re-absorbable materials

Etching with strong acidsDouble-acid etching

It is also possible to find new combined techniques that involve sanding and acid etching or sandblasting and thermal etching. The smooth implants can be electropolished or machined, the former having a surface that is subjected to an electrochemical treatment by immersion in electrolytic solution. The implants with a machined surface have a surface that appears shiny and smooth and shows streaks. Other surfaces are treated with titanium powders. The problem with this technique is the bad control of contamination and the possibility of the detachment of particles from the metal surface. There are also surfaces covered with hydroxyapatite, the latter binds to the patient’s bone and does not induce toxic or inflammatory phenomena. The sandblasted and etched surfaces, defined as SLA, are surfaces with coarse-grained and acid-etched sand. SLA surfaces have a larger contact surface than those the roughest Plasma-Spray. There are also surfaces coated with biologically active glass; experimentation on these surfaces has shown positive characteristics. The glass material is against resorption and degradation with complete replacement by the bone tissue. These surfaces are characterized by a high wettability. The purity of the surface sand the absence of contaminants is a much debated element that influences the quality and the cost of the material itself [27,28].

## 3. Results

The most available osteosynthesis material for basal condylar fractures made possible an application of plates of height 19 ± 6 mm, width 13 ± 4 mm, fixed by 4 ± 1 screw in ramus and 3 ± 1 screw in condyle. Performed tests revealed that such fixations produce 1 millimeter displacement in fracture line as 8 ± 5N force was loaded. As far as the force required for 1 millimeter displacement in fracture line after osteosynthesis was considered (Figure 3), then the six best designs were 20, 23, 10, 13, 18, and 22 and the six worst designs were 14, 11, 28, 8, 21, and 2 (Kruskal Wallis statistics = 179.77; *p* < 0.05).

Observed incidents of pull the screw out from the condyle fragment (Kruskal Wallis statistics = 1.81; *p* = 0.178) or ramus fragment (Kruskal–Wallis statistics = 0.001; *p* = 0.976) were not related to loaded force, but the number of pull out condyle screws (Chi-Square statistic = 142.4; *p* < 0.05), and ramus screws was related to the design (Chi-Square statistic = 121.7; *p* < 0.05). The least condyle screws were lost in plate designs 1, 2, 5, 6, 7, 8, 9, 10, 11, 13, 21, 22, 23, 24, and 28, the least ramus screws was observed in plate designs 1, 5, 12, 21, 22, 23, and 25. The loss of condyle screws was related to ramus screws pull out (Chi-Square statistic = 15.4; *p* < 0.05). Number of all applied fixing screws was directly proportional to force required for 1 millimeter displacement of ostesynthesized bone fragments (Kruskal Wallis statistics = 65.7; *p* < 0.05). The best results was observed as 7, 8, or 9 screws fix the plate (Figure 4). The highest forces could be bear by osteosynthesis as 4(11 ± 5N), 5(8 ± 3N), or 6 screws (10 ± 4N) were applied in ramus/distal fragment (Kruskal Wallis statistics = 62.3; *p* < 0.05). Fixation was significantly weak as 2 screws were used there (5 ± 2N). As far as number of screws in condyle portion, was considered then 4- or 3-screw fixations in at proximal fragment (10 ± 4N and 9 ± 4N, respectively, is required for 1 millimeter displacement in fracture line) were significantly better than 2-screw fixation (5 ± 2N).

The force causing 1 millimeter displacement in fracture line after fixation depended on dimensions of used plate directly proportional: on height (correlation coefficient = 0.35, R-squared = 13%, *p* < 0.05), on width (correlation coefficient = 0.52, R-squared = 27%, *p* < 0.05), and plate surface area (correlation coefficient = 0.58, R-squared = 35%, *p* < 0.05).

In this study, only one factor has been extracted during factor analysis, since only one factor had an eigenvalue greater than or equal to 1.0 (3.04). It accounted for 76% of the variability in the original data. The factor has the equation:Plate Design Factor = 0.850954 × Height (mm) + 0.846751 × Width (mm) + 0.936732 ×late surface area (mm^2^) + 0.848039 × Total fixing screws(1)
where the values of the variables in the equation are standardized by subtracting their means and dividing by their standard deviations. It also shows the estimated communalities which can be interpreted as estimating the proportion of the variability in each variable attributable to the extracted factor. Thus, one natural number describes the plate design. Design characterized by higher value of Plate Design Factor (PDF) required higher force for displacement the fixed bone fragments (moderately strong relationship between the Fmax/dL and Plate Design Factor (PDF), cc = 0.58, R^2^ = 34%, *p* < 0.05). Moreover, the factor construction causes that each plate design is significantly different from another (Figure 5, Kruskal Wallis statistic = 209, *p* < 0.05). Less screws were lost from distal/ramus fragment as Plate Design Factor higher (Kruskal–Wallis statistics = 13.4, *p* < 0.05) contrary to proximal/condyle fragment where higher values of Plate Design Factor (PDF) were related to screws pull out (Kruskal Wallis statistics = 18.2, *p* < 0.05)

The correct prediction by the neural network of condyle screws being pulled out was noted for 83% of mechanical tests and 90% pull out of ramus screws were registerd (Figure 6).

## 4. Discussion

### 4.1. Plates Combination

Two straight plates fixing the bone along the stress lines in condylar region of mandible, lead to the very rigid internal fixations. It was confirmed in many previous studies [4,6,7,8,9,11,29].

As one performs the fixation of fracture of condylar base, 3 screws in proximal fragment should use and minimum 4 screws in distal fragment (or 5–6 screws). It obviously depends on chosen design of plate in order to maintain the osteosynthesis balanced and rigid. Application of 2 screws in proximal fragment with 2 or 3 screws in distal fragment (i.e., 4- or 5-hole plates) resulted as the weakest fixations of basal condylar fractures. Plate dimensions are some related feature of the dedicated plate, and easily can be noticed that as the plate bigger, the force required to 1-millimeter displacement in fracture line higher (*p* < 0.05). The correlation coefficient equals 0.58, indicating a moderately strong relationship between the plate surface area and the displacing force, points to the valuable feature of plates design which plays significant role in stable osteosynthesis (*p* < 0.05). The highest surface area are presented by plates (390-538 mm^2^): short “A” shape condylar plate ACP (design 25), tall ACP (design 23), universal “X” shape condylar plate XCP (design 22), universal XCP with 3 + 5 hole configuration (design 19), side-dedicated XCP with 3 + 5 hole configuration (design 18), new endoscopic KLS Martin plate (design 16), big ACP (design 12) and side-dedicated XCP (design 10).

Some studies [4,13] show that the same osteosynthesis plates has been screwed in different positions and surprisingly the biomechanical effect was not worse than those of the positions suggested by the manufacturer.

Unfortunately many of presented plates design are not available nowadays but it does not mean that they had bad design. What is more they have innovative designs but without efficient delivery and sale system, the plates are not available on the market. Rigidity of plates is of paramount importance for every inventor, manufacturer, user or doctor for ORIF [open reduction and rigid fixation]. Apart from the design authors of the study wanted to describe physical properties of the condylar plates (grade of titanium alloy, annealing process, Young module, etc.), and asked manufacturers by phone and e-mails, however, majority of the manufacturers did not answer our requests for information.

Our study has not only been written directly doctors/surgeons but also inventors, constructors and medical designers. Authors want to emphasis the importance of design and present first worldwide comparison of all known designs of plates. Obviously, it is still the open question: how annealing the alloy, may it should be better to use zirconium-molybdenum alloys or titanium-niobium alloys, is 0.9 mm thickness enough or is 1.3 mm safer and/or more rigid? Those fascinating questions should be answered in the future.

### 4.2. Bioresorbable Materials

Despite the biocompatibility of titanium, many authors recommend removal for different reasons, such as metallosis, corrosion, thermal dysaesthesia, difficulties with future radiological diagnosis, malpositioning, and the migration of osteosynthesis material, particularly in craniofacial surgery [16,30]. On the other hand, bioresorbable osteosynthesis devices offer numerous advantages over metallic implants and recently systems using bioresorbable devices have been accepted as suitable tools for osteosynthesis [31]. Bioresorbable materials disappear gradually and therefore, obviate the need for removal [32]. The bending moduli of bioresorbable materials are close to that of bone and will enhance stress protection when bone support is no longer required. [30,32] The most commonly used bioresorbable material, poly-L-lactid (PLLA), is slowly degraded in the human body and physical stress is gradually transferred to the healing bone. It is believed that this property of PLLA screws prevents osteoporosis which is one of the main disadvantages of titanium fixation systems [30]. Although, some *in vitro* studies have reported the biomechanical stability of resorbable pins and osteosynthesis with resorbable screws, [16,17,32,33]. Some authors mentioned that resorbable screws exert lower retention forces than titanium ones, which result in a less stable fixation. In addition, besides their poor mechanical stability, biodegradable screws also have the number of limiting factors, such as difficult handling properties and time-consuming fixation [34]. There are some in vitro research describing unsintered hydroxyapatite/poly-L-lactide plates for subcondylar fractures. They proved that lateral strength of the bioresorbable plate system was sufficient whereas in the anteroposterior loading test, the load value at the initial displacements of 0.5 and 1.0 mm was significantly larger in the titanium plater than in the u-HA/PLLA bioresorbable plates; the load in the latter was approximately 80% that of the titanium plates. The u-HA/PLLA bioresorbable plates were much weaker than the titanium plates. However, proper plate placement for mandibular subcondylar fracture treatment greatly improved the strength of u-HA/PLLA bioresorbable plates [35].

### 4.3. FEM Analysis

The rehabilitative dentistry has always paid particular attention to the detailed analysis and the application of the occlusal forces, the distribution of tensional forces, and stress dissipation, as biomechanical factors influence the prosthetic success substantially. In time, several methods have been used to study the action of the functional forces on the prosthesis and on hard and soft tissues of the oral cavity. The finite element analysis, however, is a tool that allows analytically evaluating the distribution of tensional forces at every point of the surface taken as a reference, by creating a mathematical virtual model [36].

### 4.4. Plate Design Factor (PDF)

The proposed Plate Design Factor (PDF) can be simple measure for future plates design comparisons as far as the rigidity of osteosynthesis (force causing 1-millimeter displacement in fracture line) will be considered. The discrimination power of that factor is such high as even very similar plates in this study can reach significant difference from another (and can be individually considered). PNN procedure points that plates of construction described by Plate Design Factor over 300 (Figure 4) were the most resistant to screw pull-out as well displacing force. Plates 10, 13, 22 and 23 had a PDF > 300.

This study cannot reject the usefulness of the plates which were poorly sited in this comparison (low 1-millimetr displacement force, screw pull out, low Plate Design Factor). It can only be said that those plates should not be used (or used with great attention during reduction and occlusal control) for osteosynthesis of fractures of mandibular condyle base. It is possible that in higher level fractures those plate could be significantly better fitted. Next issue are this study limitations. Although the mechanical properties of the synthetic bones were similar to those of human bone, some differences were present in the structure of the materials. Specifically, synthetic bones have an almost uniform pore size, whereas human cancellous bone has a complex anatomical texture. This can affect the compression efficacy and fastening torque of the screws. The results of this study were based on a single-density synthetic bone; however, the biomechanical performance of the screws changes with the bone density environment [25]. The tests were conducted on synthetic bones with a perfect fracture gap simulated by parallel planes. Only specific types of fractures, types A and B, of the condyle head were simulated. These simulations were required to perform replicable and reliable testing. Finally, most screw loosening cases can be attributed to physiological cyclic loading during biting. Further evaluation of interfragmentary compression that simulates the cyclic loading of screws under physiological situations is necessary.

## Figures and Tables

**Figure 1 materials-12-03122-f001:**
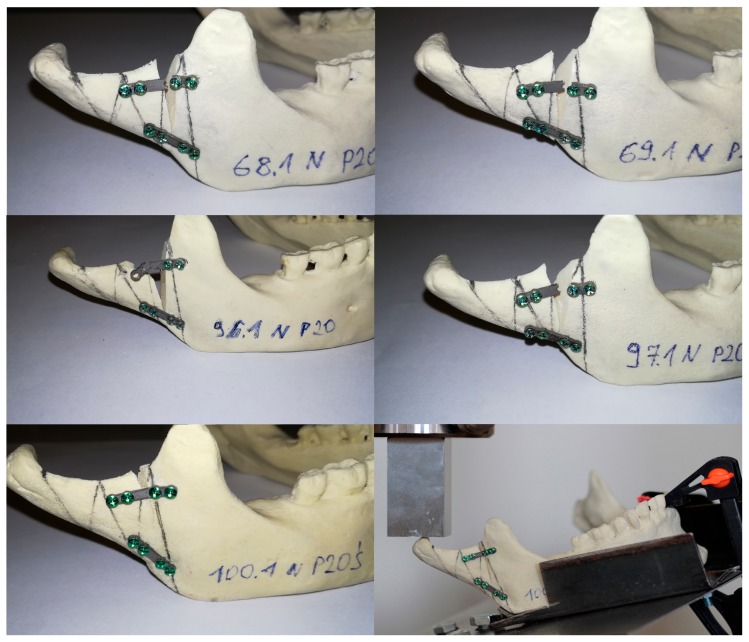
5 samples of polyurethane mandibles with broken plates number 20 and 1 sample of mandible on testing plate in the testing machine.

**Figure 2 materials-12-03122-f002:**
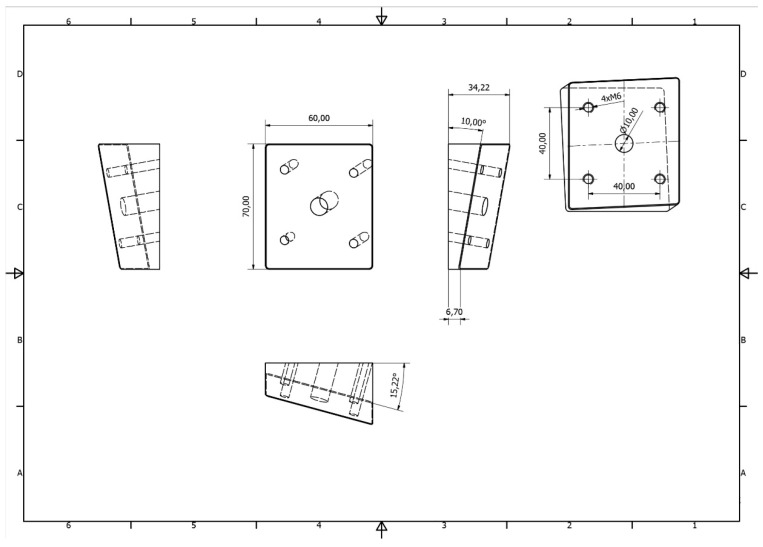
Technical drawing of individually designed stainless steel base plate for mechanical tests of mandibles.

**Figure 3 materials-12-03122-f003:**
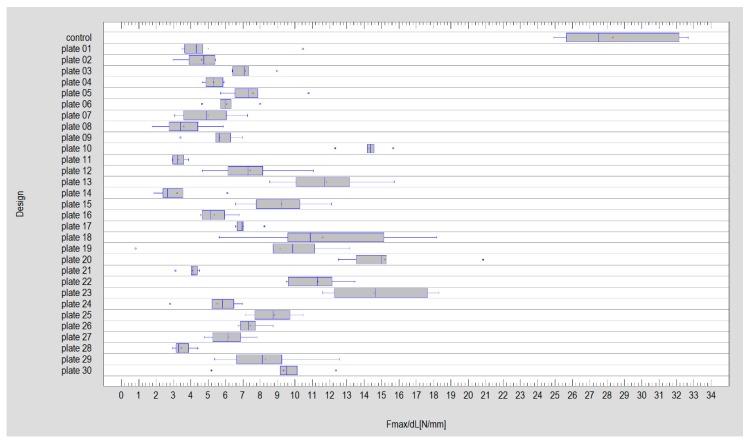
General comparison of the amount of required force for one-millimeter displacement in fracture line after plate fixation. Data in Newtons.

**Figure 4 materials-12-03122-f004:**
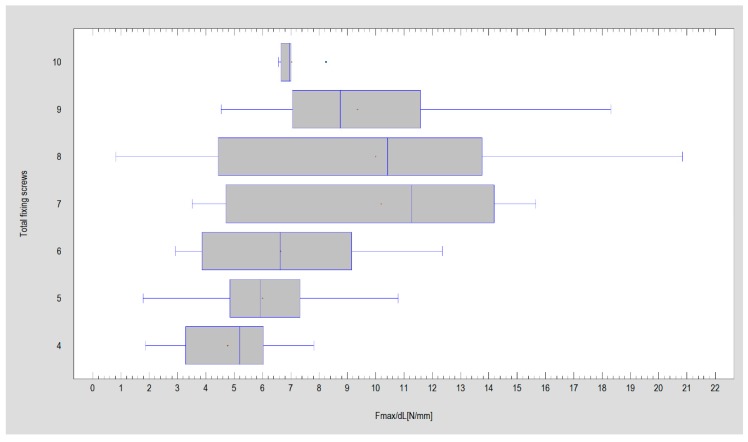
Register results of the amount of required force for one-millimeter displacement in fracture line after plate fixation in view of total number of holes designed in tested plates. The best results for plates with 7–9 holes (*p* < 0.05).

**Figure 5 materials-12-03122-f005:**
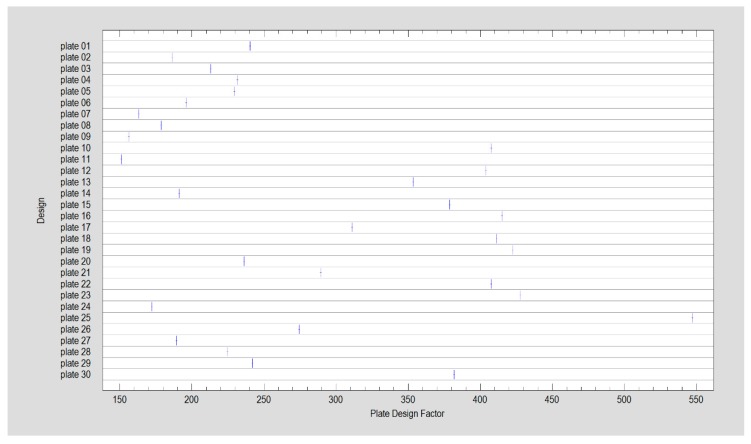
Calculated feature: Plate Design Factor, numerically describes each plate design.

**Figure 6 materials-12-03122-f006:**
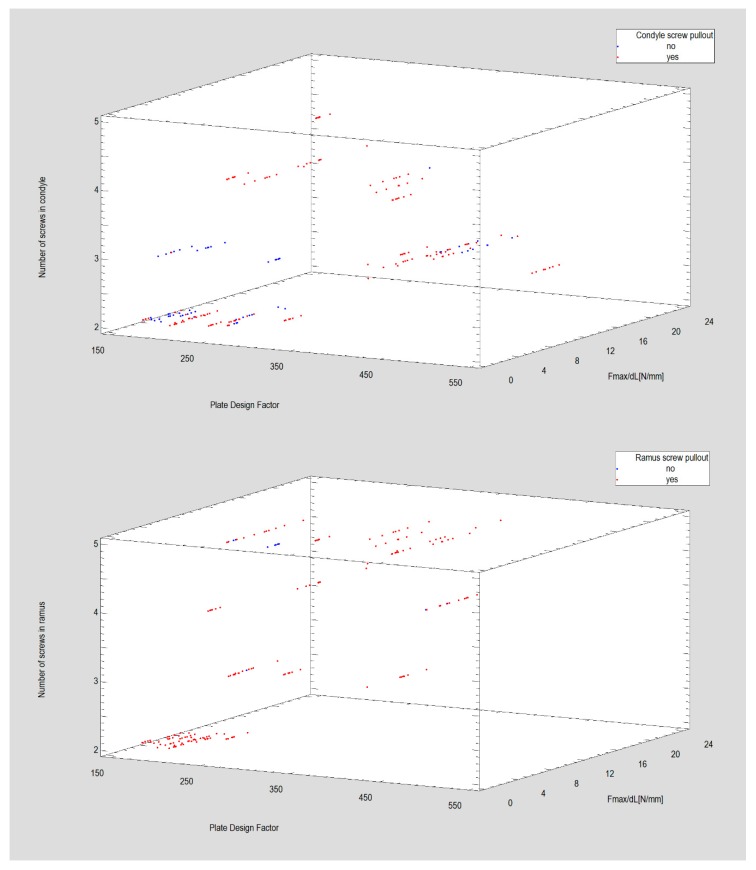
PNN procedure revealed that the incidents of condyle fragment screw loss are seldom predicted as Plate Design Factor is over 350 (plate 10, plate 13, plate 22, plate 23) in relatively high resistance to displacing forces (especially as 4 screws is fixing the plate in proximal fragment). But also, lower values (below 300) of Plate Design Factor (PDF) can predict stable plates fixed by only 2 screws in proximal fragment. Unfortunately, resistance to displacing forces in case of those plates is low. In ramus (distal fragment), the 2 screw fixational ways predicted failure, as far as pull out ramus screws parameter is considered. The best design is the plate 10 (Table 1) i.e., resistant to displacing force and described by Plate Design Factor (PDF) are over 450, and has 4-screw fixed in the ramus fragment. Moreover it can be noticed two designs described by Plate Design Factor as approx. 250 resistant to ramus screw pull out, but not so much resistant for displacing force unfortunately.

**Table 1 materials-12-03122-t001:** Tested designs of plates dedicated for osteosynthesis of basal condylar fractures of the mandible. Green cells indicate the best mechanical designs (the highest force is required for 1-millimeter displacement in fracture line after osteosynthesis, F max/dL). Red cells indicate the worst mechanical designs (the lowest force required for 1-millimeter displacement in fracture line after osteosynthesis).

**Design** **Code**	**Manufacturer** **of Similar Plate**	**Design**	**Plate Surface Area (mm^2^)**	**Plate Design Factor**	**Fmax/dL** **(N/mm)**
Plate 20	any	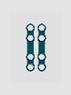	227	236	15.17 ± 2.69
Plate 03	Global D	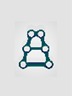	199	213	7.14 ± 0.89
Plate 29	ChM	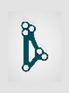	224	242	8.32 ± 2.26
Plate 11	Synthes	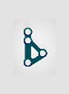	138	151	3.27 ± 0.36
Plate 06	Global D	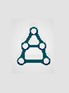	211	195	6.08 ± 1.00
Plate 14	Medartis	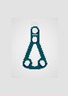	179	191	3.20 ± 1.39
Plate 08	ChM	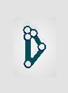	165	179	3.60 ± 1.29
Plate 24	KLS Martin	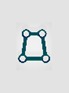	160	172	5.53 ± 1.35
Plate 07	Medartis	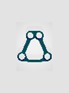	143	163	4.98 ± 1.42
Plate 09	Medartis	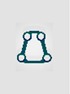	151	156	5.66 ± 1.13
Plate 27	Synthes	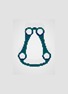	176	189	6.19 ± 1.02
Plate 02	Medartis	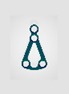	174	187	4.62 ± 0.90
Plate 26	KLS Martin	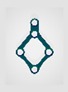	258	274	7.43 ± 0.67
Plate 05	KLS Martin	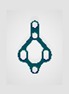	211	229	7.57 ± 1.60
Plate 04	Synthes	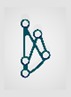	217	232	5.32 ± 0.47
Plate 21	KLS Martin	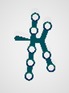	271	289	4.08 ± 0.46
**Design** **Code**	**Manufacturer** **of Similar Plate**	**Design**	**Plate Surface Area (mm^2^)**	**Plate Design Factor**	**F max/dL** **(N/mm)**
Plate 12	ChM	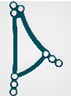	538	404	7.41 ± 1.97
Plate 13	Medartis	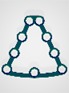	336	353	11.91 ± 2.28
Plate 15	Medartis	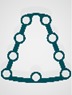	362	379	9.21 ± 1.77
Plate 01	Synthes	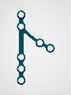	219	240	4.99 ± 2.46
Plate 28	Medicon	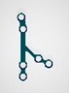	203	225	3.46 ± 0.51
Plate 17	KLS Martin	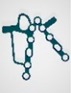	390	311	7.03 ± 0.56
Plate 16	KLS Martin	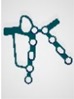	405	415	5.34 ± 0.80
Plate 25	ChM	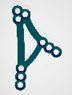	371	547	8.80 ± 1.14
Plate 23	ChM	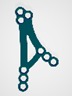	410	428	14.58 ± 2.59
Plate 30	KLS Martin	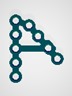	367	382	9.33 ± 2.13
Plate 22	UMed Lodz	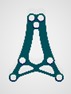	392	408	11.32 ± 1.41
Plate 10	UMed Lodz	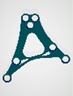	390	408	14.26 ± 0.99
Plate 18	UMed Lodz	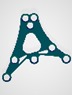	393	242	11.59 ± 4.03
Plate 19	UMed Lodz	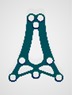	407	423	9.14 ± 3.93

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
