# Peer review of "Forces Causing One-Millimeter Displacement of Bone Fragments of Condylar Base Fractures of the Mandible after Fixation by All Available Plate Designs"

_materials, 2019, doi:10.3390/ma12193122_

Round 1

Reviewer 1 Report

Dear Authors

The study of osteosynthesis plaques is a very important topic for surgical practice. Although there are other ways to make these assessments (eg in silica methods, such as finite element analysis). This study remains of great interest.

I suggest some changes before publication:

First of all please follow MDPI guidelines for text, formatting, references style.

In introduction section please add some epidemiological information about mandible fracture and condylar basal fracture. Please briefly describe the surgical technique and used biomaterials too.
In M&M please briefly refer to plaque chemical composition and surface treatment (PMID: 30759865)
In discussion section please refer the possibility of Finite Element Analysis and Von Mises test to evaluate devices and fractures (PMID:28474002).
Thank You
Best Regards
Author Response

Dear Reviewer

Thank you for your review. I added in introduction according to your suggestion some epidemiological information and surgical procedure about mandibule fractures:

1.1. Epidemiological information

The mandible is the most vulnerable bone to fractures in the maxillofacial complex. In Europe mandibular fractures amounted to 42% of all maxillofacial fractures in a recent prevalence study. Condyle extra-capsular fractures Consisted of 26% of mandibular fractures in the same previously mentioned European study, ranking first among all types of mandibular fractures [Boffano et al., 2015]. The mandibular condyle or subcondylar region is one of the most common sites of mandibular fracture encountered, occurring between 25% and 35% of all mandibular fractures [Ellis et Throckmorton, 2005; De Riu et al., 2001].

1.2. Surgical procedures

Surgical treatment is performed under general anaesthesia. Generally maxillofacial surgeons use three different surgical approaches to reach the fracture in the region of the condylar process of the mandible. For cases where submandibular approach is necessary for load bearing osteosynthesis of associated mandibular body fractures such as in cases of atrophic mandibles the already necessitated submandibular approach was used for open reduction and internal fixation of the condylar process fracture. In cases of mild and moderately displaced condylar base fractures a transoral endoscopically assisted approach is chosen. Condylar process fractures are reached by retromandibular transparotid approaches.

In discussion we added, according to your request, some information about biomaterials:

"Despite the biocompatibility of titanium, many authors recommend removal for different reasons, such as metallosis, corrosion, thermal dysaesthesia, difficulties with future radiological diagnosis, malpositioning, and the migration of osteosynthesis material, particularly In craniofacial burgery (Schneider et al., 2012; Singh et al., 2013). On the other hand, bioresorbable osteosynthesis devices offer numerous advantages over metallic implants and recently systems using bioresorbable devices have been accepted as suitable tools for osteosynthesis. (Oki et al., 2006) Bioresorbable materials disappear gradually and therefore, obviate the need for removal. (Suuronen et al., 1991)The bending moduli of bioresorbable materials are close to that of bone and will enhance stress protection when bone support is no longer required. (Singh et al., 2013; Suuronen et al. 1991) The most commonly used bioresorbable material, PLLA, is slowly degraded in the human body and physical stress is gradually transferred to the healing bone. It is believed that this property of PLLA screws prevents osteoporosis which is one of the main disadvantages of titanium fixation systems. (Singh et al., 2013) Although, some in vitro studies have reported the biomechanical stability of resorbable pins and osteosynthesis with resorbable screws, (Wang et al., 2013; Schneider et al., 2011, 2012; Pilling et al., 2007). Some authors mentioned that resorbable screws exert lower retention forces than titanium ones, which result in a less stable fixation. In addition,

besides their poor mechanical stability, biodegradable screws also have the number of limiting factors, such as difficult handling properties and time‑consuming fixation. (Pilling et al., 2007)"

I added some information in M&M about surface treatment based on the article - PMID: 30759865. 

In M&M please briefly refer to plaque chemical composition and surface treatment (PMID: 30759865)

I added some information in Discussion about FEM based on the article - PMID:28474002

In discussion section please refer the possibility of Finite Element Analysis and Von Mises test to evaluate devices and fracturesPMID:28474002). 

I attach new manuscript.

Thank You
Best Regards

Reviewer 2 Report

I commend the authors for their efforts in testing the mechanical properties of various bone plates available commercially. The reserach strategy is flawed. Only design of the commercially available plates were used instead of industrial products. Concluding the mechanical properties using just design type is illogical. I strongly suggest testing the product itself, rather than fabricating, and compare the strengths.

Attached is a word document with other major concerns listed. 

I would suggest reconsidering after major revisions.

Author Response

Dear Reviewer   Thank you for your reviews. The method of comparing commercially available plate is fully thoughtful. Unfortunately many of presented plates design are not available nowadays but it does not mean that they had bad design. What is more they have innovative designs but without efficient delivery and sale system, the plates are not available on the market.   Rigidity of plates is of paramount importance for every inventor, manufacturer, user or doctor for ORIF [open reduction and rigid fixation]. Our study has not only been written directly to doctors/surgeons but also to inventors, constructors and medical designers. We want to emphasis the importance of design and present first worldwide comparison of all known designs of plates. All manufacturers in their factory use Ti 6Al4V for the plates we compared. Obviously, it is still the open question: how annealing the alloy, may it should be better to use zirconium-molybdenum alloys or titanium-niobium alloys, may 0.9 thickness is enough and may be 1.3 is safer/rigid. Those fascinating questions should be answered in the future.   Thank you for your review.

Round 2

Reviewer 1 Report

Dear Authors,

Thank You for Your reply. Your manuscript has been improved and it is suitable for publication now.

Kind Regards

Author Response

Dear Reviewer

It is pleasure for us for your approval of our study.

Regards

Rafal Zielinski

Reviewer 2 Report

Thank you for considering my comments/suggestions. I agree with you about the availability of variety in design and their efficiency in delivery. I suggest adding this information to the manuscript so that readers could get an idea.

Saying that, it would have been great to see at least a table having material properties of currently available commercial plates (better if those are compared with your results). Any manufacturing company would be glad to provide you with the details.

Like you said, annealing of the metal is a crucial step and definitely varies the mechanical strength on annealation process variation. Rigorous bench testing is usually done for the approvals of these plates before they are commercially out. Without knowing all these clauses, one can easily jump into conclusion that a particular design is mechanically weak (from your data). Which may or may not be true. So, I would suggest including the material data of commercially available plates and include a paragraph explaining why the original products were not measured due to their unavailability in the market, also talk about industrial manufacturing process, annealing and others, stating that the mechanical properties of the plates may vary due to the manufacturing process.

Author Response

Dear Reviewer

Thank you for your reply. We fully appreciate your thoughtful and deep review. However, despite the fact we asked every single manufacturer by e-mails and by phone asking local distributors of physical properties of condylar plates anyone answered us. We decided to write the following statements in discussion in the manuscript:

Unfortunately many of presented plates design are not available nowadays but it does not mean that they had bad design. What is more they have innovative designs but without efficient delivery and sale system, the plates are not available on the market. Rigidity of plates is of paramount importance for every inventor, manufacturer, user or doctor for ORIF [open reduction and rigid fixation]. Apart from the design authors of the study wanted to describe physical properties of the condylar plates (grade of titanium alloy, annealing process, Young module, etc.], however, any of the manufacturer answered to our request.
Our study has not only been written directly to doctors/surgeons but also to inventors, constructors and medical designers. Authors want to emphasis the importance of design and present first worldwide comparison of all known designs of plates. Obviously, it is still the open question: how annealing the alloy, may it should be better to use zirconium-molybdenum alloys or titanium-niobium alloys, may 0.9 thickness is enough and may be 1.3 is safer/rigid. Those fascinating questions should be answered in the future."

We also added the information in introduction according to your suggestion:

"The main aim of the study was to compare all known condylar plates. Researchers of this study assumed that unavailability a particular plate on the market does not mean the plate was worse than others. Thus authors did not compare original plates."

We strongly believe you understand our reply. Currently, we have been preparing another in vitro study for some plates for different fractures.

Yours sincerelly,

Rafal Zielinski

Round 3

Reviewer 2 Report

Dear Authors, thank you for considering my comments/suggestions and including the brief writeup. I do not have any furthers queries. I would recommend publishing the article in the current form.